# Experimental Design, Equilibrium Modeling and Kinetic Studies on the Adsorption of Methylene Blue by Adsorbent: Activated Carbon from Durian Shell Waste

**DOI:** 10.3390/ma15238566

**Published:** 2022-12-01

**Authors:** Quoc Toan Tran, Tra Huong Đo, Xuan Linh Ha, Thi Tu Anh Duong, Manh Nhuong Chu, Van Nhuong Vu, Hung Dung Chau, Thi Kim Ngan Tran, Phomthavongsy Song

**Affiliations:** 1Chemistry Faculty, Thai Nguyen University of Education, Thai Nguyen 250000, Vietnam; 2International School, Thai Nguyen University, Thai Nguyen 250000, Vietnam; 3Institute of Applied Technology and Sustainable Development, Nguyen Tat Thanh University, Ho Chi Minh 700000, Vietnam; 4Faculty of Food and Environmental Engineering, Nguyen Tat Thanh University, Ho Chi Minh 700000, Vietnam

**Keywords:** activated carbon, durian shell, methylene blue, adsorption, experimental design, equilibrium modeling, kinetic studies

## Abstract

For the first time, activated carbon from a durian shell (ACDS) activated by H_2_SO_4_ was successfully synthesized in the present study. The fabricated ACDS has a porous surface with a specific surface area of 348.0017 m^2^·g^−1^, average capillary volume of 0.153518 cm^3^·g^−1^, the average pore diameter of 4.3800 nm; ash level of 55.63%; humidity of 4.74%; density of 0.83 g·cm^−3^; an iodine index of 634 mg·g^−1^; and an isoelectric point of 6.03. Several factors affecting Methylene Blue (MB) adsorption capacity of ACDS activated carbon was investigated by the static adsorption method, revealing that the adsorption equilibrium was achieved after 90 min. The best adsorbent pH for MB is 7 and the mass/volume ratio is equal to 2.5 g·L^−1^. The MB adsorption process of ACDS activated carbon follows the Langmuir, Freundlich, Tempkin, and Elovich isotherm adsorption model, which has determined the maximum adsorption capacity for MB of ACDS as q_max_ = 57.47 mg·g^−1^. The MB adsorption process of ACDS follows the of pseudo-second-order adsorption kinetic equation. The Weber and Morris Internal Diffusion Model, the Hameed and Daud External Diffusion Model of liquids have been studied to see if the surface phase plays any role in the adsorption process. The results of thermodynamic calculation of the adsorption process show that the adsorption process is dominated by chemical adsorption and endothermic. The obtained results provide an insight for potential applications of ACDS in the treatment of water contaminated by dyes.

## 1. Introduction

Highly activated carbon is a commonly used adsorbent in practice due to well-developed pore size, the reactivity of surface functional groups, and large surface area that provides binding sites for adsorption [1]. The precursor used for activated carbon production is plant biomass activated carbon (activated carbon lignocellulose) which is a complex carbohydrate polymer with main components including cellulose, lignin, and hemicellulose. Lignocellulosic activated carbon can be found in agricultural and industrial waste such as duckweed [2], potato stalks and leaves [3], bamboo plants [4], mango leaves [5], tea residues [6,7,8], rice husk [9,10], sawdust [11], durian seeds and peel [12,13,14,15,16,17,18], neem leaves [19], moringa leaves and seeds [20,21], mangosteen peel [22,23,24,25], chitosan [26], pear fruit seeds [27] and acorn shells [28]. The activated carbon made from agricultural food waste has high economic efficiency in wastewater treatment due to abundant raw materials and low cost. The preparation of activated carbon consists of two main steps, (1) carbonization of the raw materials below 1000 °C in an inert gas atmosphere and (2) activation in the presence of a suitable oxidant. Most of the non-carbon compounds would be removed during the carbonization process, leaving the remaining carbon with a fixed mass and pore structure. The following activation step would enlarge the diameter of the small pores and form new pores to enhance the strong adsorption capacity of activated carbon. This step can be done chemically or physically. In general, in chemical activation, carbonization and activation are carried out simultaneously by chemical agents such as reducing agents and oxidizing agents. In contrast, during physical activation, carbonization of precursors occurs first, followed by their activation at high temperature in the presence of an activating agent, i.e., carbon dioxide or water vapor. It has been found that the chemical activation takes place at a lower temperature than physical activation. Impregnation of activated carbon precursors with chemical agents such as Na_2_CO_3_, KOH, and ZnCl_2_ can inhibit the formation of tar. Therefore, the growth of the porous structure is improved by chemical activation.

The wastewater from textile industries such as dyes, leather, paper, and plastics has become a great concern because of their adverse effects on the environment and human ecology. This wastewater has extremely high alkalinity, color, and content of organic substances and toxic solids due to the use of various chemicals in the production process. In addition, some dyes exhibit toxic properties when they penetrate into food and water sources and are carcinogenic upon consumption. Therefore, their removal from industrial wastewater is an urgent task to create a safe and clean water environment [29,30]. 3.7-bis (Dimethylamino)-phenothiazine-5-iumchloride, commonly known as methylene blue (MB) (C_16_H_18_N_3_SCl.3H_2_O) is a thiazine cationic dye with the aromatic heterocyclic structure (Figure 1) that is commonly used in paper, hair and textile colorants and dyes, and that can cause permanent injury to the eyes of human and animals [31,32].

Durian (Malvaceae family, *Durio* genus) is commonly grown in Southeast Asian countries such as India, and Sri Lanka. The shell is a residue that accounted for 70–85% of the durian fruit weight. In durian shells, there are three basic components: cellulose (30.92%); hemicellulose (17.99%) and lignin (7.69%) [12,13,14,15,16,17,18]. Therefore, the durian shell is a precursor for manufacturing and investigating the adsorption of metal ions [14] and dyes [12,14,15,18] in an aquatic environment. Inorganic acid solution activation is the process of breaking down the stable structure of lignincellulose materials. H^+^ ions break intramolecular and intermolecular bonds between cellulose, hemicellulose and lignin in biomass. Therefore, people often chemically activate it by acid [16]. Currently, the published works on activated carbon made from durian shells are very few; there is no research on making activated carbon from durian shells activated by acid H_2_SO_4_. No studying the physical properties, the iodine index of the activated carbon produced. When applying activated carbon made from durian peel to adsorb dyes and metal ions in aqueous medium, isothermal adsorption models have not been systematically studied, especially the adsorption model of the Elovich sub which has not been studied. The Weber and Morris internal diffusion model, and the Hameed and Daud external diffusion model of liquids, have not been studied to see if the surface phase plays any role in the adsorption process. The activation energy of the adsorption process has not been calculated. The results of the authors’ thermodynamic calculation of the adsorption process show that the adsorption process is physical and exothermic [2,3,4,5,6,7,8,9,10,11,12,13,14,15,16,17].

Vietnam’s durian output is expected to be 642.600 tons in 2022. More than 7 tons of durian shells are removed for every 10 tons, so this is an available, cheap, and low-cost source of raw materials. Therefore, take advantage of this available source of raw materials to make activated carbon for the treatment of polluted aqueous solution.

So, in this paper, we have made activated carbon from durian shells activated with H_2_SO_4_ acid to absorb MB in aqueous solution. The fabricated activated carbon was studied for morphological characteristics, surface functional groups and composition and specific surface area by scanning electron microscopy (SEM), Fourier transform infrared (FT-IR), energy-dispersive X-ray spectroscopy (EDS) and Brunauer–Emmett–Telle (BET) methods. Research on some physical properties included iodine index of coal. Some factors affecting the MB adsorption capacity of activated carbon have been studied such as pH, time, material mass, temperature, and initial concentration of MB. Simultaneously, study some isotherm adsorption models such as Langmuir, Freundlich, Tempkin, Dubinin–Radushkevich and Elovich, the Lagergren’s apparent first-order apparent adsorption kinetics equation, the Lagergren second-order apparent adsorption kinetics equation, the internal diffusion diffusion model of Weber and Morris, and the Hameed and Daud external diffusion model were studied. Calculation of thermodynamics and activation energy in MB adsorption process of fabricated activated carbon was thus performed.

## 2. Materials and Methods

### 2.1. Materials

#### 2.1.1. Synthesis of Activated Carbon from Durian Shell

Durian shells after being collected were washed with tap water and distilled water several times to remove all dirt particles, then placed in a kiln (Carbolite, Sheffield, UK, LMF4) and heated at 600 °C for 2 h. Under such anoxic conditions, activated carbon was thermally decomposed into porous carbonate activated carbon and hydrocarbon compounds. The samples were then removed, allowed to cool down at room temperature, and washed with distilled water until the pH 7. The samples were dried at 105 °C until a constant mass was achieved; they were then ground, and sieved to a uniform size of ≤1 mm. The physically activated durian shell activated carbon sample is denoted as DS.10 g of DSl; it was weighed and mixed with 150 mL of 0.1 M H_2_SO_4_ and allowed the reaction for 24 h. The activated carbon after being activated by acid was washed them with distilled water twice until the pH value reached between 6.5–7.0. Then, it was dried at 105 °C to a constant weight. The activated carbon made from durian shells after acid activation is denoted ACDS. They were stored in sealed vials and placed in a desiccator for further studies.

#### 2.1.2. Investigation of Physical Properties and Surface Characteristics of Activated Carbon DS and ACDS

The morphologies of the DS and ACDS were examined using scanning electron microscopy (SEM) on a JSM-6510LV unit (Jeol, Tokyo, Japan), and the chemical compositions of the ACDS were determined by energy-dispersive X-ray spectroscopy (EDS) on the same JSM-6510LV unit (Jeol, Tokyo, Japan). In addition, the surface functional groups of the ACDS were characterized by Fourier transform infrared spectroscopy (FT-IR) on a Neus 670 (Nicolet, Brighton, MO, USA). The specific surface area of the ACMP was determined by the BET method on a Micromeritics-3030 (Norcross, GA, USA). The SEM, EDS, and BET were conducted at the Institute of Materials, Vietnam Academy of Science and Technology (Hanoi, Vietnam). The FT-IR infrared spectroscopy was measured at the Department of Chemistry, University of Natural Sciences, Vietnam National University (Hanoi, Vietnam).

### 2.2. Methylene Blue Adsorption Study

#### 2.2.1. Investigation into the Factors Affecting MB Adsorption Capacity of ACDS Activated Carbon

Several factors that affect the MB removal process, such as the solution pH, time, the mass of ACDS, initial MB concentration and temperature, were investigated. Each experiment was performed at least 3 times under the same conditions and the results were the average of 3 replicates. Briefly, a certain amount (0.05 g) of ACDS activated carbon was introduced into each 100 mL Erlenmeyer flask, followed by the addition of MB dye. A total of 1 mol·L^−1^ NaOH and 0.1 mol·L^−1^ HCl were used to adjust the pH of MB solution. The flasks were agitated in various conditions. The experimental parameters included pH (3–10), reaction time (30–210 min), mass of the adsorbent (politics 0.01–0.125 g), initial MB concentration (50–500 mg·L^−1^), and temperature (303, 313 and 323 K). The results are shown in Table 1.

MB adsorption efficiency of ACDS activated carbon was calculated according to the following formula:(1)$$H=\frac{\left(C_{o}-\mathrm{Ce}\right)}{C_{o}}\cdot 100\%$$
in which H: adsorption efficiency; C_o_: initial solution concentration (mg·L^−1^) and C_e_: solution concentration when reaching adsorption equilibrium (mg·L^−1^).

All chemicals H_2_SO_4_, MB, NaOH, HCl used in the paper are Merck (Darmstadt, Germany), pure > 99%.

MB concentration before and after adsorption was measured on a 02 beam UV-Vis Molecular Absorption Spectrometer Model: UH5300, Hitachi (Tokyo, Japan, 2016). Measured with wavelength range from 190–1100 nm, scanning speed from 10–6000 nm/s, wavelength accuracy is ± 0.3 nm, noise < 0.0001 nm.

#### 2.2.2. Models of Adsorption Isotherm


*Langmuir Isotherm Adsorption Model*


The Langmuir adsorption isotherm has the form of a straight-line equation as follows [33]:(2)$$\frac{C_{0}}{q_{e}}=\frac{1}{q_{\max}b}+\frac{C_{e}}{q_{\max}}$$
in which: q_e_ and q_max_ are the equilibrium and maximum adsorption capacity (mg·g^−1^), respectively; b is Langmuir’s constant. The Langmuir equation is characterized by the parameter R_L_: R_L_ = 1/(1 + b·C_0_); If 0 < R_L_ < 1 then the adsorption is favorable; for R_L_ > 1 the adsorption is unfavorable and R_L_ = 1 then the adsorption is linear.


*Freundlich Isotherm Adsorption Model*


The Freundlich isotherm adsorption equation has the following form [33]:(3)$$\ln q_{e}=\ln K_{F}+\frac{1}{n}.\ln C_{e}$$
where: K_F_: Freundlich adsorption constant; n: Constant that is always greater than 1.


*Dubinin–Radushkevich (D-R) Isotherm Adsorption Model*


The linear form of the DR isothermal adsorption model is presented by the following equation [33]:Lnq_e_ = lnq_m_ – β·ε^2^(4)
where: β is the coefficient of adsorption energy; ε is the Polanyi potential, described as follows:(5)$$\varepsilon =\mathrm{RTln}\left(1+\frac{1}{C_{e}}\right)$$

E: Adsorption energy (kJ·mol^−1^), which can be calculated from D-R according to parameter β as follows:(6)$$E=\frac{1}{\sqrt{2\cdot \beta }}$$


*Tempkin Isotherm Adsorption Model*


The Tempkin adsorption isotherm is represented by the following equation [33]:q_e_ = B·lnK_T_ + B·lnC_e_(7)
where: B = RT/b_T_; K_T_ is the Tempkin constant.


*Elovich Isotherm Adsorption Model*


The Elovich adsorption isotherm is expressed as follows [33]:(8)$$\ln\frac{q_{e}}{C_{e}}={\mathrm{lnK}}_{e}q_{m}-q_{e}$$
where: K_e_: Elovich’s constant; q_m_: Maximum adsorption capacity Elovich.

#### 2.2.3. Investigation of Adsorption Kinetics

The Lagergren apparent first-order adsorption kinetics equation has the form [22]:(9)$$\mathrm{lg}\left(q_{e}-q_{t}\right)={\mathrm{lgq}}_{e}-\frac{k_{1}}{2.303}t$$

The Lagergren second-order apparent adsorption kinetics equation has the form [22]:(10)$$\frac{t}{q_{t}}=\frac{1}{k_{2}q_{e}^{2}}+\frac{1}{q_{e}}t$$
where: q_e_ and q_t_ are the adsorption capacity at the time of reaching equilibrium and at time t (mg·g^−1^), respectively; k_1_ and k_2_ are the first-order (time^−1^) and second-order (g·mg^−1^·time^−1^) apparent adsorption rate constants, respectively.

The internal diffusion model of Weber and Morris, derived from the Fick 2 diffusion law, was also applied to analyze the adsorption kinetics. The intraparticle diffusion model of Weber and Morris is described by the following equation [23,24]:q_t_ = K _dif_ t^0.5^ + C_i_(11)

The Hameed and Daud external diffusion model of the liquid is analyzed to investigate whether the surface phase plays a role in the adsorption process. The Hameed and Daud membrane diffusion model of the liquid is described in the form of the following equation [23,24]:ln(1−F) = −K_fd_.t(12)
where K_dif_ and K_fd_ are the intracellular and membrane diffusion rates, respectively, F represents the fraction of solute adsorbed at time t (min): F = q_t_/q_e_ is the fraction reached equilibrium equal.

#### 2.2.4. Activation Energy

To determine the activation energy according to the adsorption kinetics, h must first be determined according to the formula [34].
(13)$$h=\frac{1}{\;b}$$
where: b is the coefficient of the second order kinematics equation; k is the rate constant of the second order reaction. From that, we can calculate the activation energy according to the formula:E = RT(lnh − lnk)(14)
where: R is the gas constant (R = 8.314.10^−3^ kJ·mol^−1^·K^−1^); T is the temperature (K).

#### 2.2.5. MB Adsorption Thermodynamics of ACDS

The isobaric potential (∆G^0^), enthalpy (∆H^0^) and entropy (∆S^0^) variations of the adsorption process were calculated using the following equations [25]:(15)$$K_{D}=\frac{q_{e}}{C_{\mathrm{cb}}}$$
∆G^0^ = −RTlnK_D_(16)
(17)$${\mathrm{lnK}}_{D}=-\frac{{\Delta G}^{0}}{\mathrm{RT}}=-\frac{{\Delta H}^{0}}{\mathrm{RT}}+\frac{{\Delta S}^{0}}{R}$$
where K_D_ is the equilibrium constant.

### 2.3. Some Physical Parameters of Coal

#### 2.3.1. Determination of Humidity

ACDS activated carbon (10 g) was weighed and dried at 105 °C for about 4 h to constant weight, and then the moisture content was determined. The final result is the average value of 3 replicates [35]. Humidity is calculated according to the formula:(18)$$\%\;\mathrm{Moisture}\;\mathrm{content}=\frac{m_{o}-m}{m_{o}}\times 100$$
where: m_o_ is the initial mass of activated carbon, m is the mass of activated carbon after drying.

#### 2.3.2. Determination of the Ash Ratio

The porcelain cup was heated in the oven to constant mass, then cooled and weighed as m_1_. 1.00 g of ACDS activated carbon (m_1_) (dried at 80 °C for 24 h) was placed into a beaker and weighed as m_2_. The sample was heated in an oven at 600 °C for 4 h. The sample was cooled to room temperature and reweighed (m_3_). The ash content of ACDS activated carbon is determined as follows [35].
(19)$$\%\;\mathrm{Ash}\;\mathrm{content}=\frac{m_{3}-m_{1}}{m_{2}-{\;m}_{1}}\times 100$$

#### 2.3.3. Determination of the Density

10 g (m) of ACDS activated carbon (dried at 100 °C for 4 h) was weighed and transferred to a cylinder containing 200 mL of distilled water. The density of water change was determined using the following equation:(20)$$\mathrm{Density}=\frac{m}{V}$$

### 2.4. Iodine Index Measurement

Iodine index is a basic parameter used to evaluate the adsorption capacity of activated carbon. It is a measure of the average pore size content of ACDS activated carbon by iodine adsorption in solution. The average capillary size is responsible for changing the specific surface area of the activated carbon produced during acid activation.

Meanwhile, the iodine value was measured by mixing 0.5 g of ACDS with 50 mL of 0.1 N of iodine solution for 15 min. The filtrate (10 mL) was then titrated with 0.1 N of sodium thiosulfate solution and starch as an indicator. The amount of iodine in the solution was calculated according to the formula [35]:(21)$$\mathrm{Iodine}\;\mathrm{number}\;(\mathrm{mg}\cdot g^{-1})=\frac{\left(V_{1}N_{1}-V_{2}N_{2}\right)}{W}126.93_{f_{g}}$$
where V_1_ is the analyzed iodine volume (mL); V_2_ is the volume of Na_2_S_2_O_3_ used (mL); W is activated carbon weight (g); N_1_ and N_2_ are the iodine and Na_2_S_2_O_3_ normality (N) respectively; f_p_ is the dilution factor; and 126.93 is the iodine amount corresponding to 1 mL of Na_2_S_2_O_3_ solution.

### 2.5. Determination of the Isoelectric Point of ACDS

0.1M NaCl solutions was prepared with the initial pH (pH_i_) adjusted in increments from 0.94 to 8.35. A total of 9 conical flasks of 100 mL capacity were placed into the flask containing 0.05 g ACDS. Then, the NaCl solution with increasing pH_i_ was added to the 100 mL conical flasks and allowed to react for 48 h. Afterwards, the solution was filtered and the pH (pH_f_) determined. The difference between initial pH (pHi) and equilibrium pH (pH_f_) is ΔpH = pH_i_ − pH_f_. Plot represents the dependence of ΔpH on pH_i_, and the point of intersection of the curve with the coordinates at which the value ΔpH=0 gives the isoelectric point to be determined [34].

## 3. Results and Discussion

### 3.1. Investigation of Physical Properties and Surface Characteristics of Activated Carbon DS and ACDS

SEM imaging results of activated carbon DS and ACDS are presented in Figure 2A,B, respectively.

As can be seen from Figure 2B, the durian shell after being activated with H_2_SO_4_ has a markedly changed surface morphology, as compared to the inactivated one. Specifically, before activation, the durian shell has a rougher and less porous surface (Figure 2A). After adsorbents were activated with H_2_SO_4_, the tubular capillaries were formed, creating more voids, increasing porosity on the surface, and thus indicating the potential applications of these chemicals as adsorbents (Figure 2B). This can be explained by the fact that the activation process with H_2_SO_4_ accelerated the carbon burning process, thereby breaking down the lignocellulose in the durian shells accompanied by the evaporation of volatile compounds containing carbon in the form of CO and CO_2_ porous, thereby establishing holes on the surface of the sample.

The results of EDS analysis of ACDS activated carbon are shown in Table 2 and Figure 3. ACDS has a high carbon content by mass (82.05%) and by atom (87.06%) and low oxygen content, by mass (15.09%) and by atom (12.02%). This shows that ACDS activated carbon has effective adsorption capacity in removing dyes, heavy metal ions and other organic pollutants in water environment [20].

The functional groups on the surface of ACDS were examined through FT-IR infrared spectroscopy in Figure 4A. Results have shown that there is a broad-spectrum band at 3449.26 cm^−1^ representing the -OH group, and the spectral pattern at frequency 1625.38 cm^−1^ is assigned as the C=O carbonyl group. The IF absorption of the symmetric CH_3_ group was shown at the 1384.74 cm^−1^ spectral band, the 1060.14 cm^−1^ band can be attributed to the valence vibration of the C-O of the carboxylic acid (-COOH) group [22]. The results of FT-IR infrared spectroscopy analysis show that they are consistent with the results of EDS analysis.

The surface properties and capillary structures of ACDS activated carbon were studied by an N_2_ adsorption–desorption method and the isothermal adsorption curve was shown in Figure 4B. The nitrogen of ACDS activated carbon has a type IV form and an H4 hysteresis loop. Type IV isotherms are common in porous adsorbents, where capillary condensation occurs [22]. Furthermore, the H4 hysteresis loop is commonly found in the adsorbents with the particle sizes ranging from microcapillary to medium pore sizes. Since the nitrogen adsorption–desorption isotherm is observed at the relative pressure (P/P_0_) from 0.40 to 0.90, it is related to the adsorption of medium capillary size particles, while the curve detected when P/P_0_ is greater than 0.90 related to the adsorption of microcapillary-sized particles is less observed [22]. Thus, ACDS activated carbon mainly has average pore size. On the other hand, the results from BET measurement show that the average capillary volume is 0.153518 cm^3^·g^−1^ and the average pore diameter is 4.3800 nm, which is consistent with the results of the nitrogen adsorption–desorption isotherm. ACDS activated carbons have a specific surface area of 348.0017 m^2^·g^−1^. This shows that activated carbon made from durian peels activated with H_2_SO_4_ acid gives a larger surface area than H_3_PO_4_ acid activation of 257.50 m^2^·g^−1^ [36]. This is similar to the results of the research group of Athiwat Sirimuangjinda and colleagues when making activated carbon from scrap tires via hydrochloric acid and sulfuric acid [37]. Overall, above analysis results show that ACDS activated carbon can be considered as a potential adsorbent.

### 3.2. ACDS Isoelectric Point

The results of the isoelectric point determination of ACDS are shown in Figure 5. The isoelectric point $\left({\mathrm{pH}}_{\mathrm{pzc}}\right)$ of ACDS was determined as ${\mathrm{pH}}_{\mathrm{pzc}}$= 6.03. This shows that when pH < ${\mathrm{pH}}_{\mathrm{pzc}}$, the ACDS surface is positively charged, and when pH > ${\mathrm{pH}}_{\mathrm{pzc}},$ the ACDS surface is negatively charged.

### 3.3. Results of Physical Parameters and Iodine Index of ACDS

The physical parameters of ACDS coal are shown in Table 3. The ash content indicates the mineral content of the coal, and the hygroscopic nature can affect the adsorption capacity. It can be seen that, since ACDS has low moisture content and density, this is a favorable point for ACDS in terms of mechanical properties, as the high moisture content would reduce the strength and durability of the coal and alter the thermal engineering properties. The iodine index, which indicates the adsorption capacity as well as the porosity of ACDS, is 634 mg·g^−1^. This result is within the range of activated carbon that is suitable for practical application (i.e., 500–1.200 mg·g^−1^). However, if compared with activated carbon made from mangosteen peel [35], the ash ratio, humidity and Iodine index are all lower. This shows that there is a need for improvements in the ACDS coal preparation process.

### 3.4. The Effects of Several Operation Factors on MB Adsorption Capacity of ACDS

#### 3.4.1. Effect of pH

The adsorption process is significantly affected by the pH of the medium. The change in the medium pH leads to a change in the nature of the adsorbent, the surface functional groups, the redox potential, and the form of existence of that compound. Therefore, as pH is the most important factor that affects all water treatment processes today, determining a pH range to achieve the highest efficiency is indispensable.

From Figure 6A, it can be seen that as the pH increases, both the adsorption efficiency and adsorption capacity would increase. However, such an increase was sharp and within the pH range of 3–7, then became relatively unchanged at a pH range of 7 to 10. This can be explained as when the pH value < pH_pzc_, the ACDS surface is positively charged due to the adsorption of H^+^ ions. Therefore, a repulsive force occurs between the cationic dye MB and the adsorbent surface. In addition, at lower pH concentrations, large H^+^ competitive adsorption occurs with the positively charged MB cationic dyes at the adsorption centers. Therefore, at low pH value, the adsorption efficiency is low. In contrast, when the pH > pH_pzc_, the ACDS surface becomes negatively charged due to OH^-^ adsorption, thus establishing that electrostatic attraction between negatively charged ACDS and positively charged MB dye. Therefore, at high pH values, the adsorption efficiency is large. Therefore, the best adsorption pH for MB solution was selected as around pH 7. The results of studying the effect of pH on the MB adsorption capacity of ACDS are similar to the results of Zang Zhe when studying the MB adsorption of activated carbon made from mangosteen peel [22].

#### 3.4.2. Effect of Time to Obtain Adsorption Equilibrium

The sorbent utilization efficiency depends on the absorption rate of the solute from the liquid phase to the solid phase, as assessed by the adsorption efficiency, which was measured over different time periods. As shown in the Figure 6B, the MB absorption efficiency was increased proportionally to the surveyed time. Specifically, in the period from 30–210 min, the relative absorption efficiency increases rapidly from 30–90 min. There remains relatively little change during the period of 90–210 min, as, at this point, the adsorption process has reached equilibrium. Therefore, the equilibrium time by absorption was selected as 90 min.

#### 3.4.3. Effect of ACDS Mass

The Figure 7A indicates that increasing ACDS mass could also increase the MB adsorption efficiency. This increase is more linear in mass than active gravitational ACDS, which can be explained by the enhanced surface areas and the number of the sub, and the reduced absorption is due to the capacity to mass absorption ratio. However, although the volume of ACDS increased from 0.075 to 0.125 g, the adsorption efficiency was not significantly increased from 94.95 to 97.22%. Therefore, the ACDS weight of 0.05 g and the mass/volume ratio of 2.5 g·L^−1^ were selected for the subsequent experiments.

#### 3.4.4. Effects of Temperature

As shown in Figure 7B, both adsorption efficiency and adsorption capacity were increased with increasing temperature. This can be explained that, as adsorption is an endothermic process, when the temperature is increased, the adsorption equilibrium shifts to the right direction, which reduces the concentration of the adsorbent in the solution and lead to an increase in the efficiency and adsorption capacity of the adsorption process.

#### 3.4.5. Effects of Concentrations

From Figure 8, it can be seen that when increasing the adsorbent concentration, the adsorption efficiency decreases. This can be explained by the following equation:v_hp_ = k_hp_·C·(1 − θ) θ: Coverage

At low concentrations (diluted solutions), with 1 − θ= const, as C increases, the v_hp_ would increase linearly. However, this phase only occurs to a certain extent, depending on the ionic nature and adsorbents present. Afterwards, if the C value continues to rise, the increase in v_hp_ value would become insignificant.

### 3.5. Model of Adsorption Isotherm

#### 3.5.1. Langmuir Isotherm Adsorption Model

Langmuir isotherm accounts for the surface coverage by balancing the relative rates of adsorption and desorption (dynamic equilibrium). Adsorption is proportional to the fraction of the surface of the adsorbent that is open, while desorption is proportional to the fraction of the adsorbent surface that is covered with Langmuir.

As shown in Figure 9A, the maximum adsorption capacity q_max_ = 57.47 mg·g^−1^ and b constant = 0.062 (L·g^−1^). As demonstrated in Figure 9B, the R_L_ values were calculated in a range from 0.23 to 0.03 which are all less than 1. Therefore, the Langmuir isotherm adsorption model is suitable for the adsorption process, and the MB adsorption on ACDS materials is a form of physical adsorption.

The MB adsorption capacity of activated carbons of different plant origin was compared and the results illustrated in Table 4 showed that the maximum adsorption capacity (q_max_) of ACDS activated carbon for MB is relatively high.

#### 3.5.2. Freundlich Isotherm Model of Adsorption

This isotherm defines the surface heterogeneity and the exponential distribution of active sites and their energies.

From the results of the experiment on the effects of the initial absorbent concentration on the adsorption capacity of activated carbon, the adsorption equilibrium was investigated according to Freundlich isotherm adsorption model. Figure 10A illustrated the MB adsorption process on ACDS adsorption activated carbon according to Freundlich isotherm model. The Freundlich adsorption constant K_F_ = 19.76 (L·mg^−1^), while the constant value *n* = 5.45. The K_F_ and n value were linearly correlated, which indicates the good adsorption capacity of activated carbon and revealed the formation of adsorption bonds between the adsorbent.

#### 3.5.3. Dubinin–Radushkevich Isotherm Adsorption Model

The Dubinin–Radushkevich (D-R) isotherm model is an experimental model used to determine the nature of the adsorption process (physical or chemical) [33]. The results presented in Figure 10B have shown that the correlation coefficient R^2^ determined by the D-R isotherm model is low (R^2^ = 0.6761), so the D-R isotherm model is not suitable for the MB adsorption process of ACDS activated carbon.

#### 3.5.4. Tempkin Isotherm Adsorption Model

The Temkin isothermal adsorption model based on the assumption that the heat of adsorption is due to the interaction between adsorbents (Temkin and Pyzhev, 1940) has also been investigated. This isothermal is applicable to chemisorption on solid adsorbent and liquid adsorbent. As shown in Figure 11A, the coefficient of determination R^2^ is 0.985. Therefore, the MB adsorption process of ACDS is consistent with the Tempkin model.

#### 3.5.5. Elovich Isotherm Model

The Elovich isotherm model is applied to the multilayer adsorption process. The results of calculation are shown in Figure 11B, with q_max_ = 8.42 mg·g^−1^. Table 5 is a summary table of constants calculated from Langmuir, Freundlich, Dubinin–Radushkevich, Tempkin, and Elovich isotherm adsorption models for ACDS activated carbon.

Table 5 shows that the coefficients of determination R^2^ of Langmuir, Freundlich, Dubinin–Radushkevich, Tempkin and Elovich models were respectively 0.9958; 0.9836; 0.6761; 0.985; 0.9823, respectively. Thus, the MB adsorption process of ACDS activated carbon follows the Langmuir, Freundlich, Tempkin, Elovich adsorption isotherm models. This shows that the adsorption occurs in both monolayers and multilayers, and under the condition that the activated carbon surface is not uniform, there is an interaction between the adsorbent and the adsorbent.

### 3.6. MB Adsorption Kinetics of ACDS

The results of kinetics of the MB adsorption process of ACDS are presented in Table 6 and Table 7. From the values of the parameters of pseudo-first-order kinetics equations in Table 5, the coefficient of determination R^2^ is in the range from 0.9386–0.9664. However, the equilibrium adsorption capacity values calculated from the kinetic equations are different from the experimental values (Figure 12A). Thus, the first-order kinetics equation is not suitable for MB adsorption by ACDS.

From Table 6 and Figure 12B, the coefficient of determination R^2^ is above 0.99. Besides, the equilibrium adsorption capacity values calculated from the kinetic equations are highly close to the experimental values. Besides, the values of the reactivity constants are approximately the same. Therefore, it can be concluded that the pseudo-second- order adsorption kinetic equation is suitable for the MB adsorption process by ACDS.

The intraparticle diffusion models of Weber and Morris shown in Figure 13A illustrated that the time-dependent q_t_ plot t^0.5^ is divided into 2 regions. Region 1 is attributed to diffusion of the adsorbent in solution to the outer surface of the adsorbent (external surface adsorption). Region 2 describes the adsorption phase, where intra-particle diffusion determines the rate of adsorption. Figure 13A also demonstrated that at high concentrations (112.40 mg·L^−1^), the graph of the dependence of qt on t^0.5^ in region 2 indicates that the equilibrium has not been achieved where the intracellular diffusion begins to slow down due to low concentration of adsorbent remaining in solution. However, with concentrations of 54.91 mg·L^−1^ and 84.65 mg·L^−1^ in region 2, the graph of q_t_ dependence on t^0.5^ was horizontal, showing that the internal diffusion process has reached equilibrium. Furthermore, the linear plots of region 2 do not approach to the origin, indicating that intra-particle diffusion is not the only factor determining the rate of adsorption [24].

As illustrated in Figure 13B, the membrane diffusion model of the liquid Ln(1-F) is linearly dependent on time, indicating that membrane diffusion also determines the adsorption rate. Therefore, the thick diffusion film surrounding the MB hinders the migration of MB from the liquid phase to the ACDS solid phase surface. Therefore, it can be concluded that the MB adsorption on ACDS activated carbon is dominated by membrane diffusion [24].

### 3.7. Activation Energy

The calculation results of the activation energy for the MB adsorption of ACDS activated carbon are present in Table 8, which showed that E < 25 KJ·mol^−1^. Thus, the MB adsorption process of ACDS is dominated by external diffusion [34]. The obtained results are also consistent with the results of the author [34] when studying the phosphate adsorption of ZnO nanomaterials fabricated by hydrothermal method.

### 3.8. MB Adsorption Thermodynamics of ACDS

The results are presented in Table 9.

The results from Table 9 shows that the obtained value of free energy change (∆G^0^) was ranged from −1.746 to −6.689 kJ·mol^−1^, while the entropy change (∆S^0^) is 0.246 kJ·mol^−1^, proving that too much ACDS MB adsorption is spontaneous. The value of enthalpy change (∆H^0^) obtained has a positive value (73.045 kJ·mol^−1^), thus the adsorption process is endothermic. If ΔH^0^ < 25 kJ·mol^−1^, Van der Waals force mainly leads to physical adsorption, whereas if ΔH^0^ = 40–200 kJ·mol^−1^ the chemical bonding force would mainly lead to chemisorption [32].

Overall, the results of the MB adsorption process of ACDS activated carbon follow the apparent second-order kinetics equation and the Tempkin and Elovich adsorption isotherm models with ∆H^0^ = 73.045 kJ·mol^−1^, indicating that the adsorption of MB on ACDS is dominated by chemisorption. The obtained results are also consistent with the results of author Zhang Zhe [22] when studying the MB adsorption capacity of activated carbon made from mangosteen peel.

## 4. Conclusions

We successfully made activated carbon from H_2_SO_4_-activated durian shells (ACDS). Methylene Blue adsorption capacity of ACDS activated carbon was investigated by static adsorption method and revealed that the adsorption equilibrium was achieved after 90 min. The best adsorbent pH for MB is 7 and the mass/volume ratio is equal to 2.5 g·L^−1^. The MB adsorption process of ACDS activated carbon follows the Langmuir, Freundlich, Tempkin, and Elovich isotherm adsorption model, which has determined the maximum adsorption capacity for MB of ACDS as q_max_ = 57.47 mg·g^−1^. The MB adsorption process of ACDS follows the pseudo-second-order adsorption kinetic equation, endothermic, physical, and chemical adsorption. The adsorption rate is governed by intra-particle and membrane diffusion.

Thus, the use of ACDS to adsorb MB dye has shown good results which can be the platform for further applications of ACDS in the treatment of wastewater.

## Figures and Tables

**Figure 1 materials-15-08566-f001:**
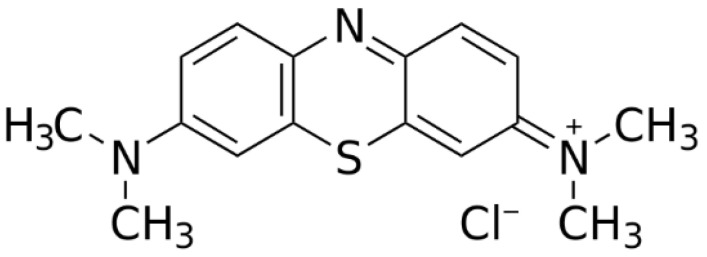
Structure of MB.

**Figure 2 materials-15-08566-f002:**
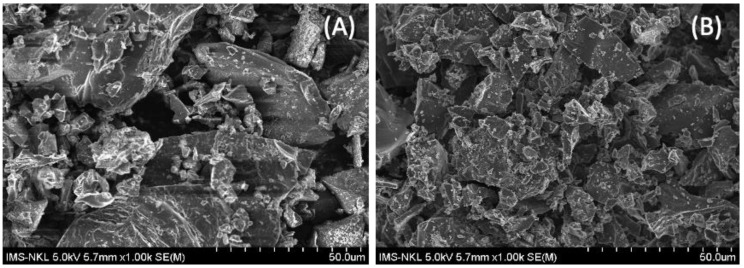
SEM image of activated carbon DS (**A**) activated carbon ACDS (**B**).

**Figure 3 materials-15-08566-f003:**
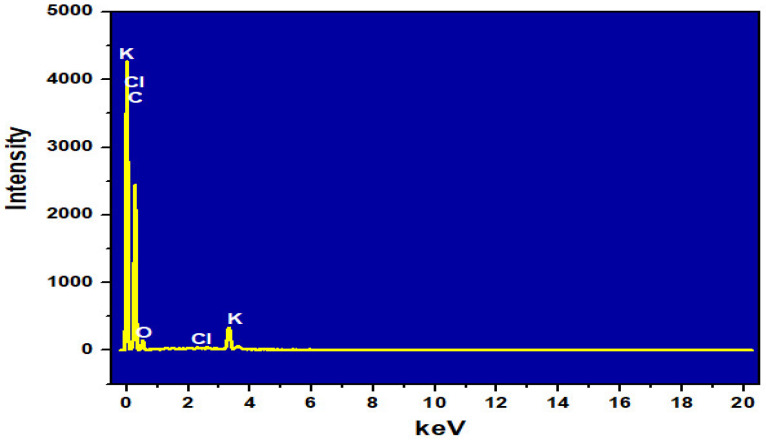
EDS spectrum of ACDS.

**Figure 4 materials-15-08566-f004:**
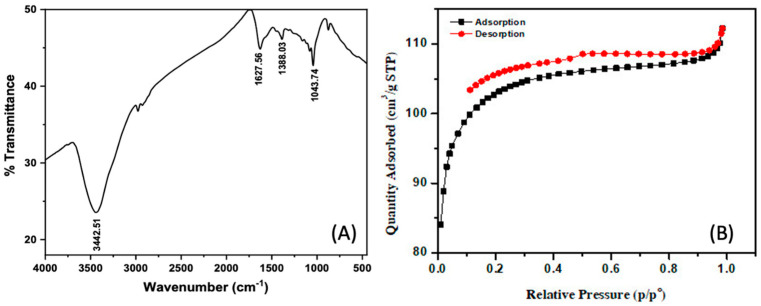
FT− IR infrared spectrum (**A**) and Nitrogen adsorption−desorption isotherms (**B**) of ACDS.

**Figure 5 materials-15-08566-f005:**
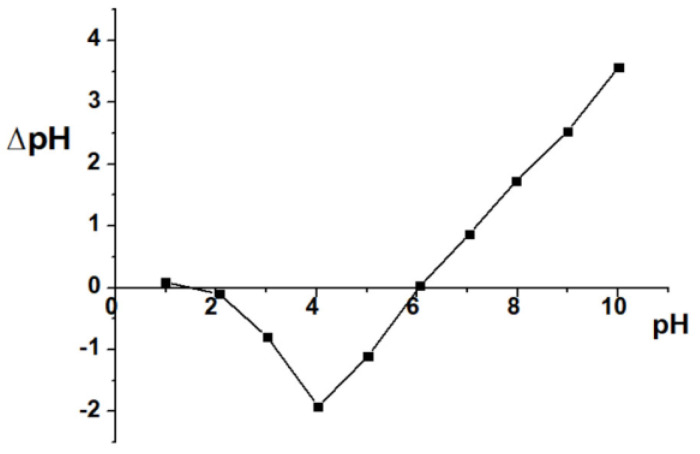
Isoelectric point of ACDS.

**Figure 6 materials-15-08566-f006:**
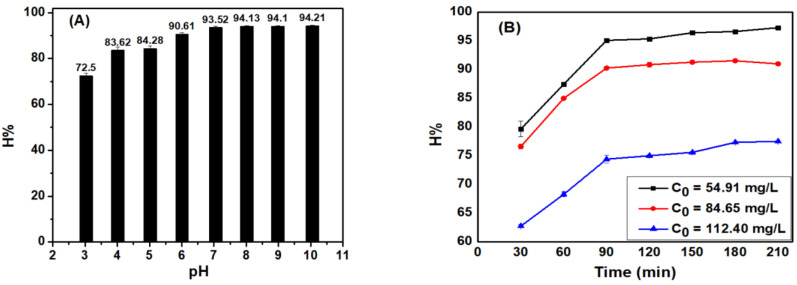
The effect of (**A**) pH and (**B**) reaction time on the efficiency of MB adsorption process of ACDS.

**Figure 7 materials-15-08566-f007:**
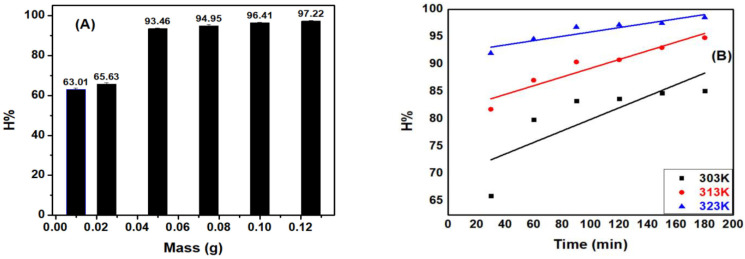
The effects of (**A**) ACDS mass on MB adsorption efficiency correlated with temperature (**B**).

**Figure 8 materials-15-08566-f008:**
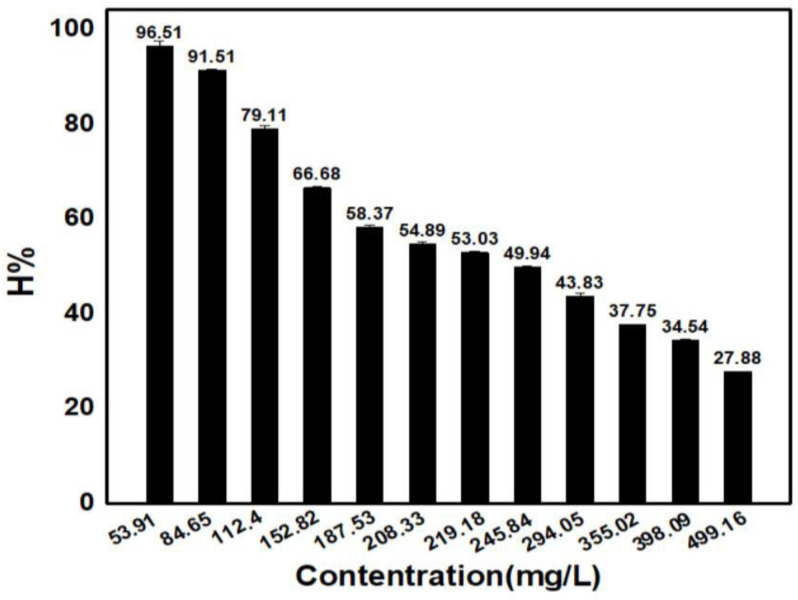
The effects of initial adsorbents concentration on MB adsorption efficiency.

**Figure 9 materials-15-08566-f009:**
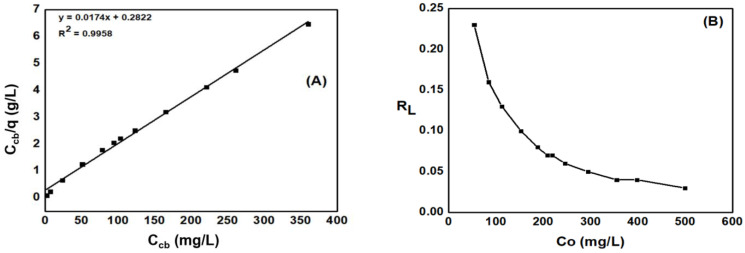
(**A**) The effect of C_cb_ on C_cb_/q for MB adsorption and (**B**) correlation of R_f_ and C_0_.

**Figure 10 materials-15-08566-f010:**
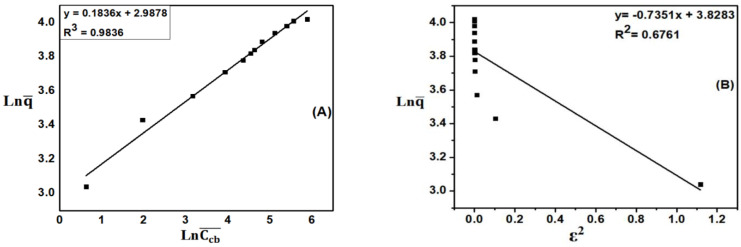
(**A**) The dependence of ln$\bar{q}$ on ln$\bar{C_{\mathrm{cb}}}$ for MB adsorption and (**B**) The dependence of lnq on ε^2^ according to Dubinin–Radushkevich isotherm model.

**Figure 11 materials-15-08566-f011:**
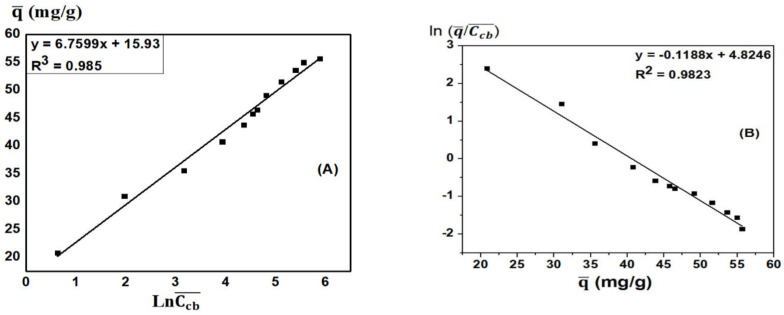
(**A**) The dependence of $\bar{q}$ on ln$\bar{C_{\mathrm{cb}}}$ according to the Tempkin model and (**B**) The dependence of ln ($\bar{q}$ /$\bar{C_{\mathrm{cb}}}$) on $\bar{q}$ according to the Elovich isotherm adsorption model.

**Figure 12 materials-15-08566-f012:**
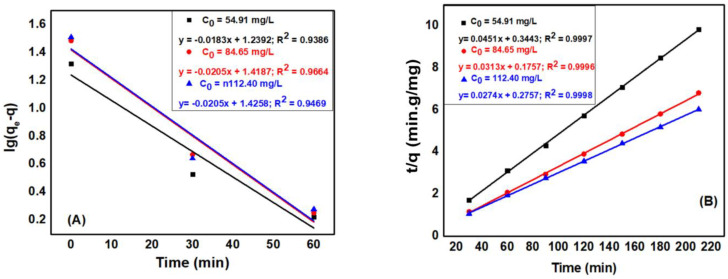
(**A**) The pseudo−first-order adsorption kinetics equation and (**B**) The pseudo-second-order adsorption kinetic equation.

**Figure 13 materials-15-08566-f013:**
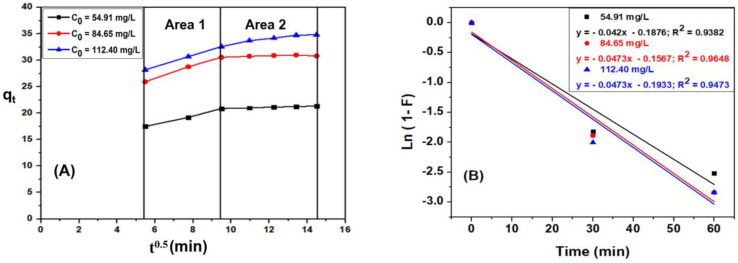
(**A**) The dependence of q_t_ on time t^0.5^ and (**B**) The dependence of ln(1 − F) on time.

**Table 1 materials-15-08566-t001:** Parameters in the expements.

Experiment/ Parameters	pH	Time (min)	Material Mass (g)	MB Initial Concentration (mg·L^−1^)	Temperature (K)
Effect of pH	3 to 10	90	0.05	50	298
Effect of time	7	30–210	0.05	50	298
Effect of mass of initial materials	7	90	0.01–0.125	50	298
Effect of MB initial concentration	7	90	0.1	50–500	298
Effect of temperature	7	90	0.1	100	303, 313, 323

**Table 2 materials-15-08566-t002:** Result of EDS analysis of activated carbon ACDS.

Elements	% Mass	% Atom
C	82.08	87.06
O	15.09	12.02
Cl	0.17	0.06
K	2.66	0.87
Total	100.00	100.00

**Table 3 materials-15-08566-t003:** Results of physical parameters, iodine index of ACDS.

	ACDS This Research	Reference [35]
Ash ratio (%)	5.63 ± 1	6.97
Humidity (%)	4.74 ± 0.03	9.8
Density (g·cm^−3^)	0.83 ± 0.05	
Iodine index (mg·g^−1^)	634 ± 0.05	1153.69

**Table 4 materials-15-08566-t004:** Maximum MB adsorption capacity (q_max_) of ACDS activated carbon and some plant biomass activated carbons.

No.	Activated Carbon	Adsorption Capacity (mg·g^−1^)	Sources
1	Neem Leaf Powder activated by ZnCl_2_	58.60	[19]
2	Tea grounds	85.16	[6]
3	Beech sawdust treated with CaCl_2_	13.02	[11]
4	Rice straw	40.58	[9]
5	Durian seeds and shell	90.90	[15]
6	Rice husk ash	8.59	[10]
7	Moringa Leaf Activated carbon	136.99	[20]
8	Durian shell	57.47	This study

**Table 5 materials-15-08566-t005:** Values of constants of Langmuir, Freundlich, Dubinin–Radushkevich, Tempkin, Elovich adsorption models for ACDS.

Isothermal Model	Parameter	
Langmuir	K_L_ (L·mg^−1^)	0.062
	q_max_ (mg·g^−1^)	57.47
	R^2^	0.9958
Freundlich	K_F_ (mg·g^−1^)·(mg·L^−1^)^1/n^	19.76
	n	5.45
	R^2^	0.9836
Dubinin–Radushkevich	q_max_ (mg·g^−1^)	45.98
	β (mol^2^·J-^2^)	−0.7351
	R^2^	0.6761
	E (kJ·mol^−1^)	0.825
Tempkin	K_T_	10.55
	b_T_ (kJ·mol)	0.367
	R^2^	0.985
Elovich	q_max_ (mg·g^−1^)	8.42
	K_e_	14.79
	R^2^	0.9823

**Table 6 materials-15-08566-t006:** Parameter values of pseudo-first-order adsorption kinetics equation.

MB Concentration (mg·L^−1^)	q_e_, Experiment (mg·g^−1^)	q_e_, Caculation (mg·g^−1^)	Constant k_1_ (min^−1^)	R^2^
54.91	20.89	17.35	0.042	0.9386
84.65	30.57	26.22	0.037	0.9664
112.40	32.60	26.66	0.057	0.9469

**Table 7 materials-15-08566-t007:** Parameter values of pseudo-second-order adsorption kinetic equation.

MB Concentration (mg·L^−1^)	q_e_, Experiment (mg·g^−1^)	q_e_, Caculation (mg·g^−1^)	Constant k_2_ (min^−1^·L·mg^−1^)	R^2^
54.91	20.89	22.22	0.0058	0.9997
84.65	30.57	31.95	0.0056	0.9996
112.40	32.60	34.50	0.0057	0.9998

**Table 8 materials-15-08566-t008:** Activation energy parameter values.

MB Concentration (mg·L^−1^)	The of Pseudo-Second-Order Adsorption Kinetic Equation	Constant k_2_ (min^−1^·L·mg^−1^)	h	E (kJ·moL^−1^)
54.91	$\frac{t}{q_{t}}$ = 0.045x + 0.3485	0.0058	2.87	15.37
84.65	$\frac{t}{q_{t}}$ = 0.0313x + 0.1764	0.0056	5.67	17.15
112.40	$\frac{t}{q_{t}}$ = 0.0274x + 0.2812	0.0027	3.56	17.80

**Table 9 materials-15-08566-t009:** Thermodynamic parameters for MB adsorption process.

T(K)	∆G^0^ (kJ·moL^−1^)	∆H^0^ (kJ·moL^−1^)	∆S^0^ ((kJ·moL^−1^·K^−1^)
303	−1.746	73.045	0.246
313	−3.456		
323	−6.689		

## Data Availability

All the data is available within the manuscript.

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
