# Peer review of "Experimental Design, Equilibrium Modeling and Kinetic Studies on the Adsorption of Methylene Blue by Adsorbent: Activated Carbon from Durian Shell Waste"

_materials, 2022, doi:10.3390/ma15238566_

Round 1

Reviewer 1 Report

Review of paper ‘Experimental Design, Equilibrium Modeling and Kinetic Studies on the Adsorption of Methylene Blue by a low-cost adsorbent: Activated Carbon from Durian Shell waste’ prepared by Quoc Toan Tran, Tra Huong Đo, Xuan Linh Ha, Thi Tu Anh Duong, Manh Nhuong Chu, Van Nhuong Vu, Chau Hung Dung, Thi Kim Ngan Tran, and Phomthavongsy Song.

The manuscript materials-1927919 is focused on the presentation of removal of dyes using a new adsorbent. I have some suggestions that authors may consider before publishing this work:

1. English needs correcting. For example, in the abstract: ‘Several factors affecting Methylene Blue (MB) adsorption capacity of ACDS activated carbon was=were investigated by static adsorption method revealed that the adsorption equilibrium was achieved after 90 min.’ or in Introduction: ‘High The activated carbon is a commonly used adsorbent in practice due its to well-developed pore size, the reactivity of surface functional groups, and large surface area that provides binding sites for adsorption [1]. The precursor used for activated carbon production is plant biomass activated carbon (activated carbon lignocellulose) which is a 40 complex carbohydrate polymer with the main components including cellulose, lignin, and hemicellulose.’

2. There are numerous editing errors in the work, including units (there is a dot instead of a multiplication note, see for example line 1, page 6), an error in the name of the compound (3,5, line 22 page 2), different font sizes in the equations, this needs to be standardised. Moreover reference numbering does not begin with 1.

3. The quality of the figures needs to be corrected. Most of them are blurred. In addition, almost every chart is made using a different technique, which creates some confusion. In Figure 13 b, move the x-scale downwards.

4 The sentence: ‘Therefore, durian shell is a precursor for manufacturing and investigating the adsorption of metal ions, dyes, and antibiotics in the aquatic environment.’ (lines 32-33, page 2) requires clarification. Please add relevant links and compare with your own results in the results section.

5. Acid activation is often used, as also mentioned by the authors. In this situation, it is important to point out why it is so important to study sulfuric acid. How does activation differ from other acids?

6. Please indicate at what lengths quantification of the dye was carried out.

7. Explain why there are differences in the drying conditions for the moisture content, ash ratio, and density measurements. What was used in the sorption tests? How do we properly refer to and compare the results when the conditions are different each time? Please refer to this in discussing Table 3

8. Choice of concentrations and pH: Does this have a bearing on real systems?

9. The symbol H is a bit misleading; in general R is used.

10. All conditions should be stated in the description of the figures (concentration of dye, pH, time amount of sorbent), especially as the authors point out differences depending on the conditions.

11. The effect of temperature on process performance should be discussed in more detail.

12. In Table 4 the authors compared the results with other natural sorbents. how does this compare with commercial sorbents?

13. Table 5: The data for adsorption isotherms were provided in individual chapters and then in the table. This needs to be modified so that the results are not duplicated in the manuscript.

Author Response

Dear Editor,

Thank you for your favorable response to our manuscript entitled: “Experimental Design, Equilibrium Modeling and Kinetic Studies on the Adsorption of Methylene Blue by a low-cost adsorbent:  Activated Carbon from Durian Shell waste”. Manuscript number: materials-1927919.

We would like to express our gratitude for the Editor and Reviewer’s efforts to improve the quality of this manuscript. We are herewith sending the revised manuscript which has been corrected based on the suggestions from you and the reviewers, including extensive revision in English grammar as well as a reorganized structure based on previous content.

We have highlighted in yellow to show changes and/or additions to the previous draft. We appreciate your time and effort in reviewing our manuscript and we look forward to hearing from you soon.

Best regards!

Reviewer 2 Report

This study used durian shells to prepare activated carbon and then tried to use it in the treatment of dye wastewater. In my opinion, this work itself is not novel because many researchers are working on similar studies. However, the strength of this study lies in its detailed work and presentation of results. I sincerely believe that the following two questions should be carefully addressed before this manuscript is considered for publication.

Please state whether the durian shells used in this work are unique? Analysis of its underlying physicochemical properties?

How does ACDS desorb the dyeing wastewater (MB dye) after adsorption? Or how is it recycled? Please give a reasonable and feasible explanation.

Author Response

(The authors gave the same response as above.)

Reviewer 3 Report

This paper describes an experimental approach to examine the removal of Methylene Blue using Activated Carbon from Durian Shell waste. In my opinion, this manuscript falls within the scope of Materials. The adsorption experiment is well designed for short scale and the manuscript is generally well organised. However, as discussed below, the methodology developed and applied here is not properly explained to be reproduced by other researchers. As a consequence, the results obtained and the conclusions drawn are arguable. I suggest a major revision of the manuscript, including additional detail and clarification regarding technical issues of the methodology and careful review and response to the comments below. Furthermore, the manuscript contains several typos and language should be significantly improved to be more concise and avoid repetition.

Title. The title should be changed since no economical-based study has been conducted herein to validate that a “low-cost” adsorbent is synthesized.

Abstract: The abstract needs more focus and needs clarity. Focus the abstract on what is new, not on general results and conclusions such as " the use of ACDS to adsorb MB dye gives good results".

In many occasions, words should be separated e.g. L.23 “theadsorption” , L. 24 “themass” …..

Introduction: In this section the authors need to point out how this study is different from the other limited literature (in brief) since Methylene Blue Dye Adsorption using Durian Shell Activated Carbon has already been studied by previous researchers e.g. Latib et al., 2013, Chandra et al,, 2007. This difference will provide the motive for the study. The concept of the manuscript is not clearly developed, and at it is not entirely clear what the novelty of the manuscript is.

Latib, E. H. A., Mustfha, M. S., Sufian, S., & Ku Shaari, K. Z. (2013). Methylene Blue Dye Adsorption to Durian Shell Activated Carbon. In Key Engineering Materials (Vols. 594–595, pp. 350–355). Trans Tech Publications, Ltd. https://doi.org/10.4028/www.scientific.net/kem.594-595.350

Thio Christine Chandra, M.M. Mirna, Y. Sudaryanto, S. Ismadji, Adsorption of basic dye onto activated carbon prepared from durian shell: Studies of adsorption equilibrium and kinetics, Chemical Engineering Journal, Volume 127, Issues 1–3, 2007, Pages 121-129, ISSN 1385-8947, https://doi.org/10.1016/j.cej.2006.09.011.

Materials and methods: I think the specific section needs major improvement.

The presentation of this crucial section seems like a guide for researchers (in an imperative way) rather than a scientific presentation of the overall process. For example, no reference to measurement conditions for SEM analysis are given or to the type of the equipment used. The authors must refer other papers on the journal as good examples to follow such a presentation manner.

I was not able to follow the overall approach and to understand the conclusions. For example, I am not even sure how many tests were conducted. There are several points that limit the scientific use of the presented methodology. In this context, crucial info are missing. Instaed general statements are provided e.g. “All chemicals H2SO4, MB, NaOH, HCl used in the paper are merk, pure > 99%.” Do the authors mean Merck? Used in this work/study?

Please provide suitable references for all models analysed and describe them in detail.

Please use uniform units e.g. not both Celsius and Kelvin.

Results and discussion. This section is premature and has a large room for improvement (major in principle).

The discussion section is rather weak. Yes, we all know that p.7 " that the activation process with 25 H2SO4 accelerated the carbon burning process, thereby breaking down the lignocellulose  in the durian shells", but arguing that " establishing holes on the surface of the sample“ is far too simple. It's much more complicated than that and therefore authors should prove increase on the surface based on the methods applied in order to answer these questions.

p.7, L.21. What do the authors mean that " Specifically, before activation, durian shell has a smooth, rough and less porous surface."? This is contradictory “rough” and “smooth”!. Please explain in detail this issue.

p.7, L.22. Please correct “aadsorbents”

Over-estimation. Authors state that " The results of EDS analysis of ACDS activated carbon are shown in Table 2 and Figure 3. ACDS has a high carbon content by mass (82.05 %) and by atom (87.06 %) and low oxygen content, by mass (15.09 %) and by atom (12.02 %). This shows that ACDS activated  carbon has effective adsorption capacity in removing dyes, heavy metal ions and other organic pollutants in water environment". This statement/conclusion is totally overestimated since EDS practically relies on several sources of errors/miss-leadings (overlapping, low energy, very low peaks etc) when are not covered by other standards or analytical characteristics tests. The reliability of a EDS quantification depends on many factors. The most critical is the composition itself. If there appears a strong overlapping of peaks the reliability is much worse than for a phase where no overlapping appears. Moreover, the visible element-specific signal is different, so that some elements (mainly heavy metals) can be described with a high accuracy in comparison to some other, like carbon and oxygen as in this study. Therefore, it is important here to try to consider other analysis (WDS, XPS, etc.).

Because the discussion is mainly based on a local absorbent, one might think that the paper is mainly for local interests. Therefore, the authors must explain more about implications for other areas. Discussion with comparison with other studies are also not enough.

Most figures are of poor quality. Please use bigger fonts, avoid overlapping between graphs or text boxes, use appropriate captions. A detailed legend should be included with every figure and thus make it stand-alone.

Other comments

-Limitation and scope of the study should be provided at the end.

-The readers may wonder what are the new contributions the present research can provide?

- Please try to avoid starting sentences and paragraphs describing Tables and Figures.

Author Response

(The authors gave the same response as above.)

Round 2

Reviewer 1 Report

I have no further comments.

Author Response

Dear Editor,

Thank you for your favorable response to our manuscript entitled: “Experimental Design, Equilibrium Modeling and Kinetic Studies on the Adsorption of Methylene Blue by a low-cost adsorbent:  Activated Carbon from Durian Shell waste”. Manuscript number: materials-1927919.

We would like to express our gratitude for the Editor and Reviewer’s efforts to improve the quality of our manuscript. We are really grateful when the article was accepted for publication.

Reviewer 3 Report

The manuscript now reads well. As such, it can be accepted for publication. 

Author Response

(The authors gave the same response as above.)
